# Decreased Nitrogen and Carbohydrate Metabolism Activity Leads to Grain Yield Reduction in Qingke Under Continuous Cropping

**DOI:** 10.3390/plants14142235

**Published:** 2025-07-19

**Authors:** Zhiqi Ma, Chaochao He, Jianxin Tan, Tao Jin, Shuijin Hua

**Affiliations:** 1Institute of Crop and Nuclear Technology Utilization, Zhejiang Academy of Agricultural Sciences, Hangzhou 310021, China; 18810153687@163.com (Z.M.); whcche@126.com (C.H.); 2State Key Laboratory of Hulless Barley and Yak Germplasm Resources and Genetic Improvement, Lhasa 850002, China; m18808010927@163.com

**Keywords:** carbohydrate, continuous cropping, metabolism, nitrogen, Qingke (*Hordeum vulgare* L. var. *nudum* Hook. f.)

## Abstract

Qingke (*Hordeum vulgare* L. var. *nudum* Hook. f.), a staple crop in the Tibetan Plateau, suffers from severe yield reduction under continuous cropping (by 38.67%), yet the underlying mechanisms remain unclear. This study systematically investigated the effects of 23-year continuous cropping (23y-CC) on the nutrient dynamics, carbohydrate metabolism, and enzymatic activities in Qingke leaves across five developmental stages (T1: seedling; T2: tillering; T3: jointing; T4: flowering; T5: filling). Compared to the control (first-year planting), 23y-CC significantly reduced leaf nitrogen (N), phosphorus (P), and potassium (K) contents by 60.94%, 47.96%, and 60.82%, respectively, at early growth stages. Key nitrogen-metabolizing enzymes, including glutamate synthase (GOGAT), glutamine synthase (GS), and nitrate reductase (NR), exhibited reduced activities under 23y-CC, indicating impaired nitrogen assimilation. Carbohydrate profiling revealed lower starch and glucose contents but higher sucrose accumulation in later stages (T4–T5) under 23y-CC, accompanied by the dysregulation of sucrose synthase (SS) and invertase activities. These findings elucidate how continuous cropping disrupts nutrient homeostasis and carbon allocation, ultimately compromising Qingke productivity. This study provides novel insights into agronomic strategies for mitigating continuous cropping obstacles in Qingke.

## 1. Introduction

The Qinghai–Tibet Plateau is considered agriculturally marginal due to its extremely high-altitude environment and low temperatures. Nevertheless, agropastoral activities have historically formed the economic foundation of the region [1]. Qingke (*Hordeum vulgare* L. var. *nudum* Hook. f.) serves as the staple crop for Tibetan populations in China, while also functioning as a crucial source of livestock feed and household fuel. As few crops have adapted to the harsh high-altitude environment, the continuous cultivation of Qingke has become a prevalent practice in the region [2]. However, long-term monoculture leads to the significant deterioration of soil properties and crop productivity, a phenomenon known as the continuous cropping obstacle [3]. Persistent challenges in Qingke continuous cultivation production systems—including frequent disease occurrence, yield forfeiture, and soil degradation—have emerged as critical scientific bottlenecks requiring immediate investigation and mitigation.

Nitrogen (N), phosphorus (P), and potassium (K) are essential macronutrients governing plant growth and yield formation. N is a structural component of proteins and nucleic acids. P regulates energy transfer via ATP and modulates photosynthetic efficiency. K enhances stomatal conductance, enzyme activation, and the phloem transport of photoassimilates [4]. In cereals, synergistic interactions among these nutrients critically determine grain filling and yield potential.

Nitrogen metabolism, a cornerstone of plant physiology, orchestrates nitrogen assimilation, remobilization, and protein biosynthesis processes indispensable for growth and yield formation. Inorganic nitrogen in soil, primarily existing as NO_3_^−^ and NH_4_^+^, is absorbed by roots and assimilated into organic nitrogen (glutamine) via the “glutamine synthase–glutamate synthase cycle”. The ammonium transporter (AMT) family genes play an important role in NH_4_^+^ absorption. This assimilated nitrogen can then be further metabolized and utilized by plants [5]. There are five key enzymes involved in nitrogen metabolism: nitrate reductase (NR), glutamine synthase (GS), aminotransferase, glutamate synthase (GOGAT), and glutamate dehydrogenase (GDH). NR is the first enzyme in the process of plant nitrogen assimilation, reducing NO_3_^−^ to NO_2_^−^, and it is also the key enzyme and rate-limiting enzyme in the whole process of nitrate nitrogen assimilation, playing a very important role in plant metabolism regulation [6]. Studies have shown that the overexpression of the *OsNR2* gene enhances nitrate uptake efficiency and increases the grain yield in rice [7]. In tomato, the high expression levels of *NRs* were associated with enhanced disease resistance, increased soluble sugar concentration, elevated organic acid, and higher vitamin C [8]. In the next step, NO_2_^−^ is converted to NH_4_^+^ by nitrite reductase (NiR) and then assimilated into organic N via GS and GOGAT [9]. There has been little research on NiR genes.

GS catalyzes the ATP-dependent conversion of glutamate to glutamine, representing a pivotal enzyme in plant nitrogen metabolism. Cruz et al. found that two Arabidopsis *GS1* family genes respond to NH_4_^+^ stimulation and improve the assimilation efficiency of nitrogen in plant roots [10]. Cai et al. found that the overexpression of *GS1* could increase the soluble protein and nitrogen content in rice [11]. These findings collectively position NR and GS as master regulators of the nitrogen economy, with profound implications for crop productivity.

GOGAT is a pivotal enzyme in the ammonia assimilation pathway, catalyzing the reductive transfer of the amide group from glutamine to α-ketoglutarate for glutamate synthesis. Based on electron donor specificity, GOGAT is classified into three isoforms: NADH-dependent (NADH-GOGAT), ferredoxin-dependent (Fd-GOGAT), and NADPH-dependent (NADPH-GOGAT) [12]. Mounting evidence underscores GOGAT’s critical role in enhancing crop productivity and nutritional quality. For instance, the knockout of *TabZIP60* in wheat elevated NADH-GOGAT activity by 25%, concomitantly increasing lateral root proliferation, spikelet number, and grain yield under field conditions [13]. Similarly, Zhang et al. demonstrated a strong positive correlation between GOGAT activity and seed yield in rapeseed (*Brassica napus*), with high-activity genotypes exhibiting an 18% higher yield than low-activity counterparts [14]. In peanut (*Arachis hypogaea*), cultivar JH5 displayed superior GOGAT activity, which correlated with 15% higher seed protein content and 22% greater pod-filling efficiency relative to standard varieties [15]. These findings collectively highlight that the coordinated activities of GS, GOGAT, and GDH govern nitrogen remobilization and dry matter partitioning, thereby determining the ultimate crop yield [16]; however, these activities remain unexplored in Qingke under continuous cropping conditions.

Plant sugars, as primary products of photosynthesis, serve dual roles as metabolic substrates and signaling molecules, orchestrating critical developmental processes from embryogenesis to reproductive transitions. Soluble sugars (e.g., sucrose, glucose, fructose) mediate carbon partitioning and osmotic regulation, while insoluble starch synthesized through tightly regulated enzymatic cascades acts as the predominant energy reservoir in chloroplasts [17]. Sugar degradation plays a vital role in nitrogen metabolism by supplying substrates for amino acid synthesis, positioning sugars as central mediators in C-N metabolic coordination and C/N ratio homeostasis in plants [9].

The main sucrose-related metabolic enzymes are sucrose synthase (SS), sucrose phosphate synthase (SPS), and invertase. SS has the dual properties of decomposing and synthesizing sucrose and plays a key role in the regulation of sucrose metabolism. SPS is the main enzyme responsible for sucrose synthesis, which uses fructose 6 phosphate (F6P) and UDP-glucose as substrates to produce sucrose-phosphate and UDP, and then uses sucrose phosphatase to irreversibly degrade the sucrose-phosphate products to produce sucrose [18]. Invertases, divided into acidic invertases (AIs) and neutral invertases (NIs), hydrolyze sucrose into glucose and fructose.

Pyruvate kinase is one of the main rate-limiting enzymes in glycolysis, catalyzing phosphoenolpyruvate and ADP to ATP and pyruvate [19]. In a study on maize, Yu et al. found that the activities of NR, GS, SPS, and SS in the ears of nitro-deficient maize were significantly reduced, resulting in shorter ears of maize and less biomass accumulation, resulting in yield reduction [20]. Dong et al. found that rice pyruvate kinase family genes are closely related to the regulation of grain yield and quality, and the deletion of pyruvate kinase genes will lead to reductions in the total starch content and protein content, resulting in reductions in the 1000-grain weight and seed setting rate [21].

Starch biosynthesis initiates with Calvin cycle-derived fructose-6-phosphate, which undergoes sequential isomerization via phosphoglucoisomerase (PGI) and phosphoglucomutase (PGM) to generate glucose-1-phosphate. ADP-glucose pyrophosphorylase (AGPase) then catalyzes the conversion of ADP-glucose—the essential precursor for amylose/amylopectin polymerization using starch synthase, starch branching enzymes (SBEs), and debranching enzymes (DBEs); eventually, starch granules are formed [17]. Previous studies have shown that the simultaneous overexpression of ADPG genes *Bt2* and *Sh2* in maize can increase the starch content by 74% and seed weight by 15% [22]. Compared with WT, the overexpression of *LSU* in wheat strains enhanced AGPase activity and increased the thousand kernel weight by 5.2–9.1% [23]. These results indicate that the overexpression of AGP can be used to increase the crop yield.

Our previous work demonstrated that 23y-CC significantly increases reactive oxygen species (ROS) accumulation while depleting the antioxidant capacity in Qingke, contributing to a yield loss [24]. We hypothesize that ROS stress suppresses the activity of key enzymes and then reduces the accumulation of nutrients and carbohydrates, ultimately leading to yield losses. Building on this, the current study integrates physiological and biochemical analyses to unravel how nutrient limitation and carbohydrate dysregulation exacerbate yield reduction under continuous cropping. By profiling N, P, and K contents, nitrogen-metabolizing enzymes, and carbohydrate dynamics, we aim to establish a holistic framework for understanding Qingke’s adaptation to long-term continuous cropping.

## 2. Results

### 2.1. Chlorophyll, N, P, and K Content in Qingke Leaf Under 23-Year Continuous Cropping (23y-CC) at Different Developmental Stages

#### 2.1.1. Chlorophyll Contents

Green plants convert atmospheric carbon dioxide, soil water, and inorganic nutrients into organic compounds and oxygen through photosynthesis. It has been found that plant chlorophyll content is non-linear and positively related to photosynthetic efficiency [25,26]. We measured the chlorophyll a (Chl a) and chlorophyll b (Chl b) content in leaves across five critical developmental stages of Qingke: seedling (T1), tillering (T2), jointing (T3), flowering (T4), and grain filling (T5). Our results revealed that both Chl a and Chl b exhibited similar temporal trends in Qingke leaves throughout these growth phases. In the control group, the chlorophyll level peaked at the T2 stage, followed by sharp declines of 6.2% and 90.98% for Chl a, and 27.90% and 88.21% for Chl b during the T3 to T4 stages, before stabilizing thereafter. In the 23y-CC treatment, the contents of Chl a and Chl b peaked at the T3 stage, and in the T4 and T5 stages, decreased by 60.82% and 92.77% and by 58.21% and 64.49%, respectively. The Chl a and b contents in the control group were significantly higher than those in the 23y-CC treatment at the T1 and T2 stages, unlike those in the T4 stage. There was no significant difference in the T3 and T5 stages between the two groups (Figure 1). The chlorophyll content of the 23y-CC treatment was lower in the T1-T3 stage, but higher in the later development stage of Qingke when compared with the control group.

#### 2.1.2. N, P, and K Contents

During the experimental period (from T1 to T5), the accumulation trends of N, P, and K exhibited a gradually declining trend. The N, P, and K levels in the 23y-CC treatment were significantly lower than those of the control. The nitrogen (N) content followed a similar trend in both groups, declined from T1 to T2, rebounded slightly by T4, and then dropped sharply at T5. The greatest difference between the control group and the 23y-CC treatment was at stage T1, at 60.94%. The N content decreased by 70.97% and 44.91% at the T5 stage in the control and 23y-CC treatment, respectively, compared with the T1 stage (Figure 2a).

The P content of the control group showed a decreasing trend from stage T1 to T5, with a 61.88% decrease in stage T5 compared to that at stage T1. The 23y-CC treatment showed an increasing trend from stage T1 to T3, which decreased after reaching the peak in T3 and decreased by 66.19% at T5. The largest difference was 1.92-fold changes at the T1 stage between the two groups, while the smallest difference was 1.11-fold changes at the T3 stage (Figure 2b).

The K content of the 23y-CC treatment was lower than that of the wild type, except at the T4 stage. The K content increased by 12.71% from the T1-T2 stage in the control group but decreased by 37.64% in the 23y-CC treatment. There was a linear decrease from T2 to T5 in the control group, decreasing by 90.83%. In the 23y-CC treatment, the K content was relatively stable at T2-T4 and declined sharply thereafter (Figure 2c).

### 2.2. Nitrogen Metabolism Enzyme Activities in Qingke Leaf Under 23y-CC at Different Developmental Stages

#### 2.2.1. Glutamate Synthase Activity

GOGAT is a key enzyme for nitrogen assimilation and cycling in plants. Both the control and 23y-CC treatments exhibited progressive declines in GOGAT activity from T1 to T5, from 1377 nmol/min/g FW (control) to 627 nmol/min/g FW (23y-CC), reaching undetectable levels by T5. GOGAT activity was higher in the control group than in the 23y-CC treatment, except at stages T3 and T5 (Figure 3a). Fd-GOGAT uses Fd as a coenzyme, which mainly exists in green tissues such as the plastid and chloroplast, and is involved in the primary absorption of nitrogen and the reabsorption of ammonia released by photorespiration [27]. In the control group, Fd-GOGAT activity showed a 5.77% increase from T1 to T2 followed by a progressive decline. In the 23y-CC treatment, there was a 45.81% decrease from T1 to T2, a slight increase from T2 to T3, and then a decline again. There was no significant difference in T1, T3, and T4 between the two groups. The activity of Fd-GOGAT in the T2 group was 2.07-fold lower than that of the control group. On the contrary, the Fd-GOGAT activity was 1.77-fold higher than in the control group at the T5 stage (Figure 3b).

#### 2.2.2. Glutamine Synthase Activity

GS is a key enzyme of nitrogen metabolism in plants that catalyzes the synthesis of glutamine from glutamate, ATP, and NH_4_^+^ [28]. The GS activity was significantly lower with the 23y-CC treatment than that with the control group at the T2 and T3 stages, while significantly higher in T4 and T5. The GS activity exhibited stage-specific patterns; the two groups both peaked at T2 and then sharply decreased by 42.62% at T3 stage, and then were undetected at T4 and T5 in the control group (<0.5 μmol/h/g). However, they decreased by 74.31%, 93.84%, and 90.85%, respectively, under the 23y-CC treatment (Figure 3c).

#### 2.2.3. Glutaminase Activity

Glutaminase (GLS), a critical enzyme in nitrogen remobilization, catalyzes the deamination of glutamine to glutamate, thereby modulating amino acid metabolism and photorespiratory flux. The GLS activity of the 23y-CC treatment was significantly higher than that of the control at T1–T2, while significantly lower at T3–T5. In the control group, GLS activity showed an upward trend from T1 to T3, reached the peak at T3, and then sharply decreased; there was a 49.37% decrease from T3 to T5. In the 23y-CC treatment, the peak occurred at the T2 stage and then decreased, with a 77.40% decrease at T5 compared with T2 (Figure 3d).

#### 2.2.4. Nitrate Reductase Activity

NR regulates the degradation of nitrate to nitrite in plants, and its activity represents the ability of nitrogen metabolism in plants. NR activity in the control group increased first and then decreased, peaked at the T4 stage, and decreased by 80.22% at the T5 stage compared with T4. However, in the 23y-CC treatment, NR activity decreased firstly from stage T1 to T2, then increased from T2 to T3, peaked at T3, and declined sharply thereafter. It went undetected at T5 and showed a decline–rise–decline pattern under the 23y-CC treatment. With the 23y-CC treatment, NR activity was significantly higher than in the control group at the T1 stage, and there was no significant difference at the T2 stage. However, it was significantly lower than with the control group at the T3-T5 stages (Figure 3e).

#### 2.2.5. Nitrite Reductase Activity

NiR, a pivotal enzyme in nitrogen assimilation, catalyzes the reduction of nitrite to ammonium, a precursor for amino acid and protein biosynthesis [8,9,10,11,12]. Except for at the T5 stage, the NiR activity of the 23y-CC treatment was significantly lower than that of the control group, which decreased by 79.71%, 55.67%, 29.77%, and 34.66%, respectively. There was an initial rise–fall pattern from T1 to T3 in the two groups. At the T3–T5 stages, the control group also showed a rise–fall pattern, while the 23y-CC treatment showed a slow upward trend (Figure 3f).

### 2.3. Sugar Content in Qingke Leaf Under 23y-CC at Different Developmental Stages

#### 2.3.1. Soluble Sugar Content

Sugars in plants mainly include sucrose, fructose, glucose, starch, and other sugars, which are very important for the yield and quality [29]. The sucrose content in stage T2 was significantly lower, while in the T4 and T5 stages, it was significantly higher than in the control group. There was no significant difference in stages T1 and T3 between the two groups. In the control group, the sucrose content increased from the T1 to T2 stage, peaking at T2, and then sharply decreased by 79.75% at the T5 stage, ultimately returning to levels comparable to T1. Under the 23y-CC treatment, the sucrose content increased from the T1 to T3 stage, and decreased thereafter by 66.90% at T5 compared with the T3 stage. The sucrose content was 1.81-fold higher in the control group than with the 23y-CC treatment at the T2 stage (Figure 4a).

After 23 years of continuous cropping, the fructose content in stages T2 and T5 was significantly lower, while that in stages T3 and T4 was significantly higher than in the control group. And there was no difference in the T1 stage between the two groups. The fructose content peaked at the T2 stage in the control group and decreased by 37.88% at the T5 stage. With the 23y-CC treatment, T1–T3 showed an increasing trend, and T3–T5 showed a decreasing trend. There was a 64.05% decrease at T5 compared with T3 (Figure 4b).

The glucose content for the 23y-CC treatment was significantly lower than that of the control group. In the control group, the glucose content increased first and peaked at the T3 stage, being 1.64-fold higher than at stage T1. The content at T5 decreased by 10.49% compared with the T3 stage. With the 23y-CC treatment, stages T1–T4 showed an upward trend, and then decreased after reaching the peak. The glucose content of T4 was 34.01-fold higher than that at T1, and then decreased by 46.74% at T5 compared with T4. The greatest disparity between the two groups was observed at the T2 stage, with a 7.30-fold difference, while the smallest divergence occurred at the T4 stage (1.35-fold difference) (Figure 4c).

#### 2.3.2. Starch Content

We measured the starch content of leaves in five stages. The results show that the starch content of the two groups followed a slow increasing trend, and at T5, the starch content had increased by 54.29% and 50.77% compared with the T1 stage, respectively. There was no difference between the two groups at T1 and T2. The starch content of the 23y-CC treatment at stages T3 to T5 was significantly lower than that of the control group (Figure 4d).

These results show that the glucose and starch contents of the 23y-CC treatment were lower than those of the control group. The sucrose content was lower than that of the control group in the early development stage and higher in the late stage. The fructose content in the early and late development stages was lower than that in the control group, but higher in the middle stages (Figure 4).

### 2.4. Sucrose Metabolism Enzyme Activity in Qingke Leaf Under 23y-CC at Different Developmental Stages

#### 2.4.1. Sucrose Synthase Activity

SS, a bifunctional enzyme central to sucrose metabolism, governs both sucrose synthesis (via SS-II) and catabolism (via SS-I), critically regulating carbon partitioning and cell wall biogenesis [30,31]. The activity of SS-II in the 23y-CC treatment was significantly higher than that in the control group, except for at stage T5. In the control group, SS-II activity showed an upward trend at the T1-T3 stages, decreased at T4, and increased again slightly at T5. SS-II activity decreased at the T1-T2 stages, increased at the T2-T3 stages, and then declined again under the 23y-CC treatment. Finally, SS-II activity decreased by 79.70% at the T5 stage compared with the T3 stage (Figure 5a).

There was a completely opposite trend from stages T1 to T5 between the two groups. The control group’s SS-I activity declined by 23.98%, while the 23y-CC treatment group showed the following: a 22.53% increase at stages T1-T2; control activity that rebounded 1.89-fold, contrasting with a 61.27% reduction in the treated group at stages T2-T3; and a progressive 61.31% decrease in controls versus 1.31-fold activation in treated samples at stages T3-T5. The 23y-CC treatment maintained significantly elevated SS-I activity at T1, T2, and T5 compared to the controls, with maximal divergence at T2 (3.48-fold). Conversely, the smallest divergence occurred at the T5 stage (1.58-fold), suggesting stage-specific inhibition or accumulation patterns (Figure 5b).

#### 2.4.2. Sucrose Phosphate Synthase Activity

Sucrose synthesis in plant photosynthetic tissues is primarily mediated by SPS [32]. Comparative analysis revealed that the 23y-CC-treated group exhibited significantly suppressed SPS activity relative to the control, with marked 2.41-, 6.88-, and 109.27-fold reductions at stages T1, T4, and T5, respectively. In the control group, SPS activity demonstrated a biphasic pattern: an initial 34.67% decrease during T1-T2, followed by a transient 1.29-fold recovery at T3, ultimately declining by 53.28% from T3 to T5. The SPS activity in the 23y-CC treatment increased, firstly peaking at the T3 stage and then decreasing, followed by complete enzymatic inactivation at T5, approaching detection limits (Figure 5c).

#### 2.4.3. Sucrose Phosphorylase Activity

Sucrose phosphorylase (SP), a member of the glycoside hydrolase family 13 (GH13), catalyzes the hydrolysis of sucrose into α-glucose-1-phosphate and fructose. The 23y-CC-treated group exhibited significantly lower SP activity compared to the control group. Both groups demonstrated peak enzymatic activity at the T3 developmental stage, with no statistically significant difference observed between stages T2 and T3. Notably, the 23y-CC treatment showed 1.29-, 1.22-, and 4.13-fold increases in SP activity at stages T1, T4, and T5, respectively, when compared to corresponding stages in the control. In the control group, SP activity maintained relative stability throughout all five experimental periods. Contrastingly, the 23y-CC-treated group displayed an initial increase followed by a subsequent decrease: a gradual 22.23% activity increase from T1 to T3 was followed by a sharp 44.13% decline from T3 to T5 (Figure 5d).

#### 2.4.4. Invertase Activity

Invertase, a critical enzyme in sucrose metabolism, catalyzes the hydrolysis of sucrose into glucose and fructose, providing substrates for plant growth and stress signaling [33,34]. Based on pH optima, invertases are classified into acid (AI, pH 4.5–5.5) and neutral (NI, pH 6.5–7.5) isoforms, with further subcellular localization distinctions including cytoplasmic (CIN), cell wall-bound (CWIN), and vacuolar (VIN) forms [35]. CWIN, in particular, mediates sucrose partitioning and stress responses [36,37,38]. The analysis of NI activity in Qingke leaves revealed a conserved U-shaped trajectory in both the control and 23y-CC-treated groups, though with temporal divergence. The control group reached minimal NI activity at T3, while the 23y-CC treatment group exhibited delayed suppression, attaining its nadir at T4. There was no significant difference in stages T1 and T5 between the two groups. The NI activity of stages T2 and T3 was significantly higher than that of the control group, while significantly lower at the T4 stage. Both groups rebounded by T5, with control and treated samples achieving 1.76- and 1.79-fold increases from their respective minima (Figure 5e).

The activity of soluble acid invertase (S-AI) decreased from T1 to T5. The control group underwent rapid initial inactivation (74.92% decrease, T1–T2), followed by gradual attenuation (84.08% decline, T2–T4) and partial T5 recovery (97.27% vs. T4). In contrast, the 23y-CC treatment group exhibited a slight S-AI reduction (30.84% T1–T3) and then a sharp decrease from T3 to T5, culminating in near-zero activity at T5. In the control group, the T5 stage exhibited a decrease of 92.13% compared with the T1 stage, while the 23y-CC treatment group exhibited a decrease of 95.95%. The S-AI activity with the 23y-CC treatment was significantly lower than that in the control group at T1, but significantly higher at T2, T3, and T4 with no difference at T5 (Figure 5f).

The cell wall insoluble acid invertase (B-AI) activity in the control group remained stable at the T1–T2 stages, followed by linear activation peaking at T4 (55.46% increase over T2) and abrupt T5 inactivation. CW-AI activity in the 23y-CC treatment group induced transient T1–T2 inhibition (16.88% decrease), partial T2-T3 recovery (38.6% increase), and complete enzymatic collapse at the T4 and T5 stages. There was no significant difference in stage T1 between the two groups, and the biggest difference was in stage T4 (Figure 5g).

### 2.5. Starch Metabolism Enzyme Activity in Qingke Leaf Under 23y-CC at Different Developmental Stages

#### 2.5.1. Pyrophosphate: Fructose-6-phosphate 1-Phosphotransferase Activity

Pyrophosphate: Fructose-6-phosphate 1-phosphotransferase (PFP) is a cytoplasmic enzyme that is widely present in plant tissues. The conversion between fructose-6-phosphate and fructose-1,6-diphosphate is reversibly catalyzed through phosphorylation and dephosphorylation [39]. In the control group, PFP activity exhibited a unimodal trajectory, peaking at the T2 stage before undergoing progressive inactivation, declining by 71.94% (T3), 93.12% (T4), and 96.35% (T5) relative to T2. By contrast, the 23y-CC treatment maintained stable PFP activity from T1 to T3, with only marginal attenuation at later stages. Comparative analysis revealed dramatic activity amplification in controls during early development, exhibiting 5.09-, 8.68-, and 2.45-fold higher PFP activity than the 23y-CC group at T1, T2, and T3, respectively. Notably, this disparity dissipated by T4–T5, with both groups converging to equivalent activity values (Figure 6a).

#### 2.5.2. Fructose-1.6-diphosphatase Activity

Fructose-1.6-diphosphatase (FBP), a key regulatory enzyme in sucrose biosynthesis, catalyzes the irreversible hydrolysis of fructose-1,6-bisphosphate to fructose-6-phosphate, bridging the Calvin cycle’s regenerative phase with starch and sucrose partitioning [40]. Both the control and 23y-CC-treated groups exhibited progressive FBP inactivation from T1 to T5, though with divergent kinetic trajectories. In controls, FBP activity declined sharply, showing 49.56% (T2), 63.03% (T3), 88.36% (T4), and 93.99% (T5) reductions relative to T1. In the 23y-CC treatment group, the activity in stages T2 to T5 decreased by 30.84%, 67.46%, 82.24%, and 94.98%, respectively, compared with the T1 stage. Cross-group comparisons revealed sustained FBP suppression in the 23y-CC group at T1–T4 (Figure 6b).

#### 2.5.3. ADP-Glucose Pyrophosphorylase Activity

AGPase is one of the main regulatory enzymes of the starch synthesis pathway, which converts glucose-1-phosphate and ATP into ADP-glucose and pyrophosphoric acid, where ADP-glucose is the basic substrate of starch synthesis [41,42]. Distinct temporal activation patterns emerged between the experimental groups: The control group displayed unimodal AGPase activity, gradually increasing from T1 to peak at T4 (3.50-fold vs. T1), followed by a 66.54% collapse at T5. In contrast, the 23y-CC treatment group exhibited oscillatory dynamics, with activity rising at T2, declining sharply at T3, rebounding at T4, and ultimately collapsing to levels at T5 that were statistically indistinguishable from those at T2. The AGPase activity of 23y-CC showed 1.87-fold higher activity than that in the control group at the T1 stage, but was significantly lower than the control group at stages T3 and T4 (Figure 6c).

#### 2.5.4. Pyruvate Phosphate Dual Kinase Activity

Pyruvate phosphate dual kinase (PPDK), a bidirectional enzyme central to phosphoenolpyruvate (PEP) metabolism, mediates the reversible interconversion of pyruvate, ATP, and inorganic phosphate (Pi) with PEP, AMP, and pyrophosphate (PPi). This reaction opposes pyruvate kinase (PK)-catalyzed PEP-to-pyruvate flux, positioning PPDK as a critical node in carbon partitioning between glycolysis and starch synthesis [21]. The 23y-CC treatment exhibited a linear PPDK activity decrease from T1 to T5, with reductions of 21.85% (T2), 38.43% (T3), 76.96% (T4), and 96.67% (T5) relative to T1. In contrast, controls displayed unimodal activation, peaking at T2 (7.01-fold increase vs. T1) before declining by 21.21% (T3), 70.52% (T4), and 95.74% (T5) from the T2 maximum. The PPDK activity in the 23y-CC treatment was significantly higher than that in the control group at the T1 stage, but significantly lower than that in the other stages (Figure 6d).

#### 2.5.5. Triose Phosphate Isomerase Activity

Triose phosphate isomerase (TPI) is a sugar metabolism enzyme that catalyzes dihydroxy acetone phosphate (DHAP) and transforms into glyceraldehyde 3-phosphate (G3P) [43]. The activity of TPI showed the same trend between the control and 23y-CC treatment groups. TPI activity of 23y-CC treatment was significantly lower than that of the control group at stages T1-T4. In the control group, TPI activity decreased by 53.78% (T2), 77.77% (T3), 78.99% (T4), and 76.60% (T5) compared with T1, while decreasing by 53.17% (T2), 74.66% (T3), 76.39% (T4), and 76.93% (T5) under the 23y-CC treatment. The difference between the two groups was largest at the T1 stage, with a 1.56-fold difference (Figure 6e).

## 3. Discussion

Continuous cropping obstacles are pervasive agricultural challenges affecting diverse crops, including staple cereals (wheat, maize), horticultural species (tomato, watermelon, cucumber), and economic crops (soybean, tobacco). They manifest through consistent physiological aberrations: stunted growth, malformed root architecture, compromised stress resilience, and diminished yield and quality [44]. Many studies have suggested that changes in soil physical and chemical properties, deterioration in the soil biological environment, and a decrease in the number and type of beneficial bacteria are the main factors causing continuous cropping obstacles [45,46,47]. In previous studies, we investigated the content of ROS and antioxidant capacity and found that ROS significantly increased, the scavenging capacity of ROS significantly weakened under the 23y-CC treatment of Qingke, and the flavonoid and tannin contents decreased under continuous cropping, which is an important reason for its low antioxidant capacity. Moreover, the stress caused by ROS finally led to the yield reduction in Qingke [24]. In this study, we hypothesized that ROS stress under continuous cropping conditions suppresses the activity of the key enzymes involved in nitrogen metabolism and sugar conversion. This inhibition significantly reduces the accumulation of nutrients and carbohydrates, ultimately leading to yield losses [48,49]. Based on this, we analyzed the changes in chlorophyll content, N, P, and K content, nitrogen metabolism enzyme activity, carbohydrate content, and sugar metabolism enzyme activity in the leaves of Qingke under continuous cropping conditions. The chlorophyll content of the 23y-CC treatment was found to be lower than that of the control group in the early development stage, but increased in the later development stage. The nutrient and carbohydrate contents in leaves were lower than in the control group, and the corresponding enzyme activity was also significantly decreased. The resultant carbohydrate deficiency during grain filling exacerbates yield penalties, highlighting carbohydrate metabolism as a key vulnerability under continuous cropping. The activity changes in GOGAT, GLS, NR, etc., imply the severe dysregulation of nitrogen metabolism, potentially disrupting photorespiration–ammonium salvage pathways, exacerbating nitrogen imbalance and yield penalties under continuous cropping [6,7,8,9,10,11,12,13,14,15,16]. In rice, the OsWRKY23-DNR1 module enhances nitrate uptake and GOGAT activity, boosting NUE in indica vs. japonica subspecies [50]. NR and GS facilitate nitrate and ammonium assimilation, while SPS promotes carbon conversion. NR activity increases under shading conditions but exacerbates C-N imbalance, reducing grain quality [51]. Regarding the AGPase rate-limiting enzyme for starch biosynthesis, low temperatures reduce AGPase activity in wheat, decreasing the amylose/amylopectin content [52]. These enzymes represent the key pathways through which plants integrate carbon and nitrogen into their metabolic networks, sustaining growth and development [53]. This metabolic gridlock—early-stage photosynthetic limitation compounded by chronic nutrient deprivation—might explain the yield penalty cascade.

Despite the advances in characterizing the phenotypic and physiological manifestations of continuous cropping obstacles in Qingke, the molecular mechanisms driving yield reduction remain inadequately resolved. These mechanisms include the identification of candidate genes and functional validation, multilayer regulation pathways, evolutionary adaptation, and so on. As a multifaceted stress syndrome, continuous cropping induces the dysregulation of gene expression networks through two synergistic pathways: the suppression of housekeeping genes essential for primary metabolism and aberrant activation of stress-responsive paralogs that divert resources from growth processes. In future research, the integrated employment of functional genomics, transcriptomics, proteomics, and metabolomics technologies will enable comprehensive analysis of transcriptional and translational alterations under continuous cropping conditions. This multi-omics approach will elucidate the critical gene regulatory networks and metabolic compounds driving the pathogenesis of continuous cropping disorders. Notably, epigenetic mechanisms—particularly histone acetylation and methylation modifications—exert crucial regulatory control over plant stress responsiveness. Systematic epigenomic profiling can effectively identify candidate genes and their associated signaling cascades implicated in monoculture-induced physiological dysfunction. The strategic integration of these molecular discoveries with innovative agronomic interventions constitutes a pivotal advancement toward resolving persistent yield limitations in continuous cropping systems.

Plant autotoxicity constitutes a critical biological mechanism underlying continuous cropping obstacles, defined as the phytotoxic effects induced by root exudates and decomposing residues of conspecific plants through the release of bioactive metabolites [54,55]. In continuous cropping systems, the progressive accumulation of these autotoxic compounds, particularly phenolic acids and terpenoids, disrupts soil microbiome equilibrium and impairs key physiological processes (e.g., mitochondrial electron transport, auxin signaling), ultimately inhibiting subsequent crop establishment and growth [56]. The important allelopathic products in higher plants are divided into simple water-soluble organic acids, straight-chain aliphatic alcohols, fatty aldehydes and ketones, and fatty alcohols. Higher plants produce diverse allelochemicals categorized into 15 major classes: hydrophilic compounds: water-soluble organic acids (e.g., malic, citric acids), straight-chain alcohols, and aliphatic aldehydes/ketones; phenolic derivatives: simple phenols, benzoic acids (e.g., salicylic acid), cinnamic acids (e.g., ferulic acid), coumarins, and tannins; isoprenoids: mono/diterpenoids and steroidal compounds; nitrogen-containing metabolites: amino acids, peptides, alkaloids (e.g., caffeine), and cyanogenic glycosides; and specialized conjugates: quinones, flavonoids (e.g., glycosylated thiocyanates), unsaturated lactones, purine analogs, and nucleoside derivatives [57]. Empirical studies across species such as Panax ginseng have demonstrated temporal intensification of autotoxicity correlated with prolonged monoculture duration [58]. Investigating the role of Qingke root exudates in mediating continuous cropping disorders might offer novel mechanistic insights into the autotoxic feedback loops underlying continuous cropping-induced yield decline, thereby establishing an innovative framework for elucidating the pathogenesis of these agricultural constraints in Qingke systems.

## 4. Materials and Methods

### 4.1. Plant Materials and Experiments

An experiment was conducted at the Tibet Academy of Agricultural and Animal Husbandry Sciences (29°56′ N, 91°07′ E; altitude 3794 m), which was the same field as in our previous study [24]. The Qingke variety Zangqing 2000 was used in the study. The experiment comprised 5 treatments including 0-, 3-, 5-, 10-, and 23-year continuous cropping. The 0-year continuous cropping was selected as a control. The 23y-CC treatment was started from 2000, the previous crop was Qingke in 1999, and no crop was planted before 1999. The 0-, 3-, 5-, and 10-year continuous cropping systems were established from 2022, 2013, 2018, and 2020, respectively, and the previous crop was rapeseed/pea because a rapeseed–pea rotation is another tillage mode in Xizang. In the current study, samples of Qingke leaves were collected in 2022. In the field trial, each block was planted with either Qingke and the pea/rapeseed rotation starting at different years, until the end of the study in 2022. The experiment employed a randomized complete block design (RCBD) with three replicates, where each block area was 100 m^2^. In the experimental durations, previous covers were removed from each plot. Qingke seeds were manually sown at a seeding rate of 225 kg ha^−1^ during 25th to 30th in March under each treatment. For fertilizer application, nitrogen (N), phosphorus (P_2_O_5_), and potassium (K_2_O) fertilizers were applied at rates of 22 kg ha^−1^, 10 kg ha^−1^, and 13 kg ha^−1^, respectively, from the commencement of continuous cropping treatment. Furthermore, the fertilizer amounts were kept the same in each continuous cropping treatment. During rapeseed and pea rotation, the nitrogen (N), phosphorus (P_2_O_5_), and potassium (K_2_O) fertilizers were applied at rates of 12 kg ha^−1^, 8 kg ha^−1^, and 10 kg ha^−1^, respectively, for rapeseed growth and at the rates of 12 kg ha^−1^, 7 kg ha^−1^, and 9 kg ha^−1^ for pea growth. There was no irrigation during the experiment. The soil was slightly alkaline (control: pH = 8.42; 23y-CC: pH = 8.01). Because we did not collect soil samples before the continuous cropping treatment, the soil nutrient content was estimated at the end of the Qingke growth period in 2022. The soil total N, P, and K contents in the control were 0.91, 0.55, and 0.46 g 100 g^−1^ and in the 23y-CC treatment were 1.08, 0.38, and 0.18 g 100 g^−1^. The temperature and precipitation conditions during the growth period are described in a previous study [24]. Qingke leaves were collected at five critical developmental stages: seedling (T1), tillering (T2), jointing (T3), flowering (T4), and grain filling (T5) for the 0y-, 5y-, and 23y-CC treatments but not all the treatments because of the large quantity of work. To mitigate edge effects, sampling was strictly confined to the central plot areas in one third of the sampling area [24]. A part of the samples was flash-frozen in liquid nitrogen and stored at −80 °C for enzyme activity assays; fresh samples were inactivated at 100 °C for 30 min and then dried at 70 °C for the determination of some physiological indices.

### 4.2. Chlorophyll Content Quantification

Leaf chlorophyll extraction and quantification were performed using commercial assay kits (Keming Bio-Tech, Suzhou, China) following the manufacturer’s protocols for Qingke leaf tissues: Weigh 0.1 g of fresh leaf tissue and cut it into small pieces. Add 1 mL of distilled water and 50 mg of reagent one. Grind thoroughly under weak light conditions. Transfer the slurry to a 10 mL volumetric flask and add extraction solvent to the 10 mL flask. Incubate the slurry in the dark for 3 h until complete chlorophyll bleaching. Measure supernatant A663/A645 zero with the solvent blank. Calculate the content using a formula.

### 4.3. Leaf Nutrient Content Analysis

The leaf nitrogen concentration was determined using the micro-Kjeldahl method with modifications [59]. Phosphorus quantification followed the molybdenum blue method [60]: Mix 100 μL of the sample with 900 μL of the molybdate reagent, and incubate at 45 °C for 20 min. Add 200 μL of ascorbic acid, vortex immediately, develop the color at 75 °C for 15 min, and measure the A_820_ against the reagent blank. Extract potassium with 1 M NH_4_OAC (pH 7.0), and determine the potassium content via flame photometry calibrated with KCl standards [61].

### 4.4. Carbohydrate Profiling

The sucrose, fructose, glucose, and starch content of Qingke leaves was determined using commercial kits (Keming Bio-Tech, Suzhou, China) following the manufacturer’s protocols: Weigh 0.1 g of fresh leaf tissue and grind it in a mortar. Add the corresponding reagents and follow the procedure in the manufacturer’s instructions. After centrifugation, collect the supernatant for subsequent analysis. Measure the absorbance and calculate the concentration according to the manufacturer’s protocol.

### 4.5. Enzyme Activity Assays

The enzyme activities of Qingke leaves were determined using commercial kits (Keming Bio-Tech, Suzhou, China) following the manufacturer’s protocols: Weigh 0.1 g of fresh leaf tissue, add 1 mL of ice-cold extraction buffer, and homogenize on ice. Centrifuge at 10,000× *g* at 4 °C for 10 min; take the supernatant, and place it on ice for testing. Measure the absorbance and calculate the concentration according to the manufacturer’s protocol.

### 4.6. Statistical Analysis

In this study, we only provide the data of the 0- and 23-year continuous cropping treatment because the results of the 5y-CC treatment showed no significant influences on the crop growth under the 5y-CC treatment. Data are expressed as mean ± SD (*n* = three biological replicates). Significance analysis with a two-tailed Student’s *t*-test was implemented in Microsoft Excel 2016. Significance thresholds: * *p* < 0.05.

## 5. Conclusions

Our previous investigation showed that the continuous cropping of Qingke resulted in a yield reduction, partly due to the weakened ROS sequestration during plant growth and development. In the current study, we systematically evaluated the impacts of continuous cropping on nutrient and photosynthate accumulation, partitioning, and their metabolism in plants. Macro-nutrients, including N, P, and K, significantly decreased under the 23y-CC treatment compared to that in the control. The decline in N content was caused by the reduction in the activities in N assimilation and of transformation enzymes. However, both the sucrose and starch contents were significantly lower at early and late developmental stages under the 23y-CC treatment compared to the control. Furthermore, the glucose content under the control was significantly higher than that under the 23y-CC treatment due to the stronger catalyzation of sucrose by SPS and all the enzymes involved in starch metabolism. These results indicate that the nutrient and carbohydrate deposition and metabolism represent key factors contributing to impaired plant growth and development during the continuous cropping of Qingke. In the future, it is necessary to alter the cropping system in Qingke, such as through rotation with different crops. Rotation between leguminous and cereal crops helps regulate the soil organic matter content, restore and improve the soil structure, enhance the fertility, and ultimately increase the yield [62,63]. The mixed cropping system of Qingke and pea increases both the grain and forage yield simultaneously [64]. Based on these previous studies, we hypothesize that intercropping Qingke with pea could effectively mitigate the continuous cropping barrier of Qingke.

## Figures and Tables

**Figure 1 plants-14-02235-f001:**
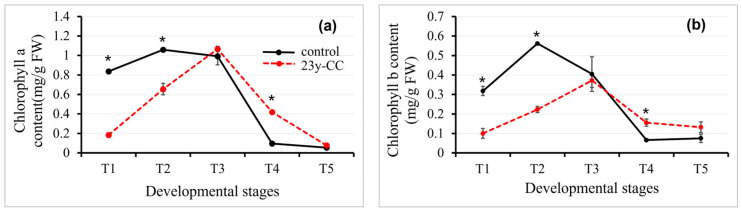
Chlorophyll a (**a**) and chlorophyll b (**b**) content in Qingke leaves from stages T1 to T5 between the control and 23y-CC. T1–T5 indicate the seedling, tillering, jointing, flowering, and seed-filling stages, respectively. Data represent means ± SD (*n* = three biological replicates). Asterisks indicate statistical significance based on two-tailed Student’s *t* tests (* *p* < 0.05). FW: fresh weight.

**Figure 2 plants-14-02235-f002:**
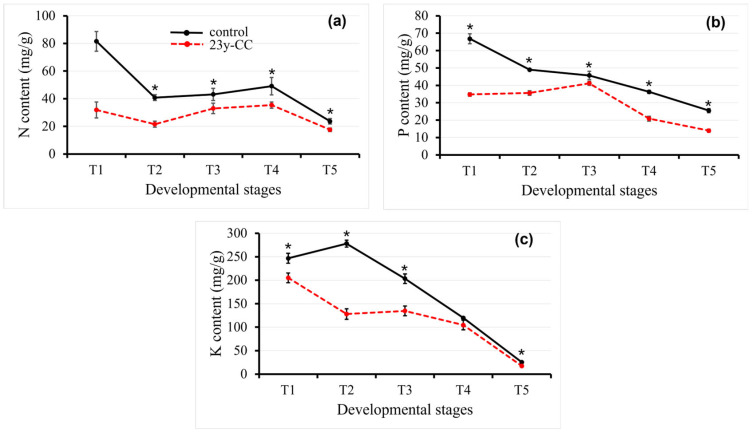
The N (**a**), P (**b**), and K (**c**) content in Qingke leaves from stages T1 to T5 in the control and 23y-CC treatment. T1–T5 indicate the seedling, tillering, jointing, flowering, and seed-filling stages, respectively. Data represent means ± SD (*n* = three biological replicates). Asterisks indicate statistical significance based on two-tailed Student’s *t* tests (* *p* < 0.05).

**Figure 3 plants-14-02235-f003:**
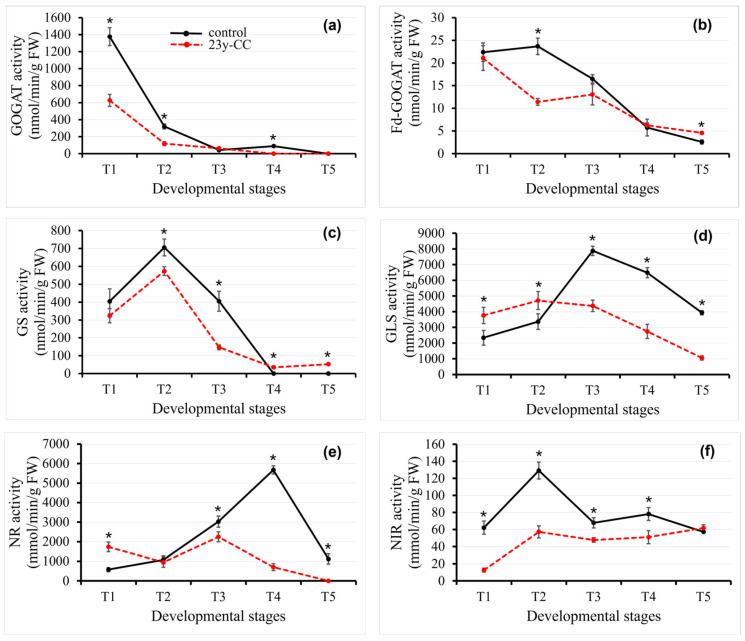
GOGAT (**a**), Fd-GOGAT (**b**), GS (**c**), GLS (**d**), NR (**e**), and NiR (**f**) activity in Qingke leaves from stages T1 to T5 between the control and 23y-CC. T1–T5 indicate the seedling, tillering, jointing, flowering, and seed-filling stages, respectively. Data represent means ± SD (*n* = three biological replicates). Asterisks indicate statistical significance based on two-tailed Student’s *t* tests (* *p* < 0.05). FW: fresh weight.

**Figure 4 plants-14-02235-f004:**
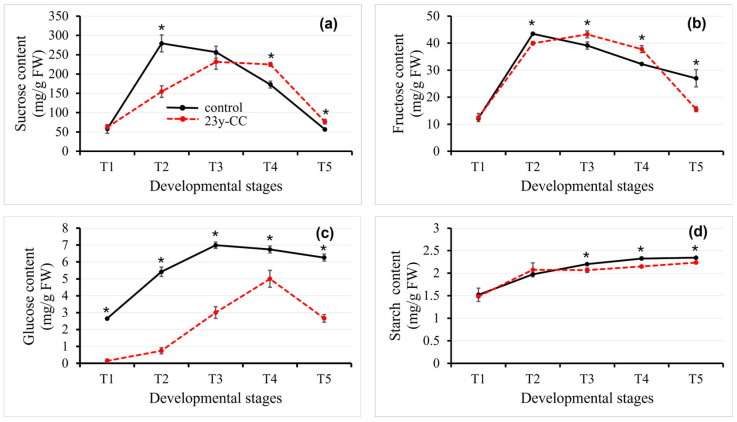
Sucrose (**a**), fructose (**b**), glucose (**c**), and starch (**d**) content in Qingke leaves from stages T1 to T5 between the control and 23y-CC groups. T1–T5 indicate the seedling, tillering, jointing, flowering, and seed-filling stages, respectively. Data represent means ± SD (*n* = three biological replicates). Asterisks indicate statistical significance based on two-tailed Student’s *t* tests (* *p* < 0.05). FW: fresh weight.

**Figure 5 plants-14-02235-f005:**
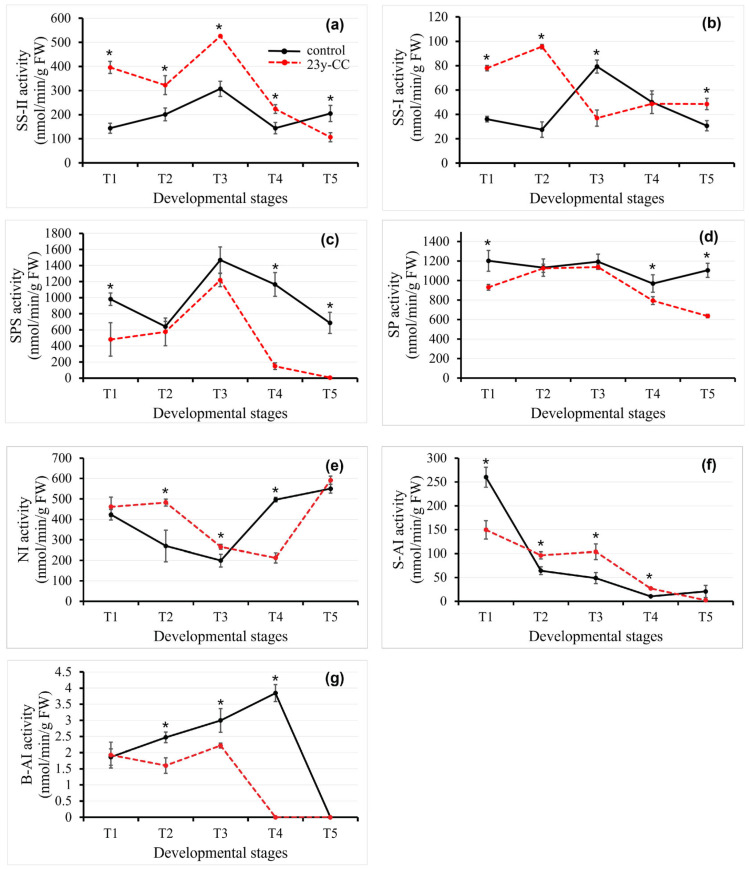
SS-II (**a**), SS-I (**b**), SPS (**c**), SP (**d**), NI (**e**), S-AI (**f**), and B-AI (**g**) activity in Qingke leaves from stages T1 to T5 between the control and 23y-CC groups. T1–T5 indicate the seedling, tillering, jointing, flowering, and seed-filling stages, respectively. Data represent means ± SD (*n* = three biological replicates). Asterisks indicate statistical significance based on two-tailed Student’s *t* tests (* *p* < 0.05). FW: fresh weight.

**Figure 6 plants-14-02235-f006:**
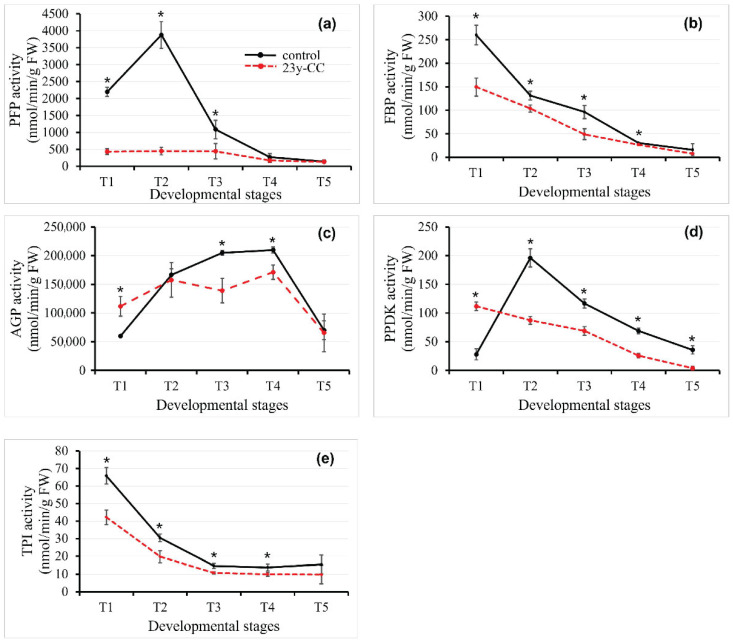
PFP (**a**), FBP (**b**), AGP (**c**), PPDK (**d**), and TPI (**e**) activity in Qingke leaves from stages T1 to T5 between the control and 23y-CC groups. T1–T5 indicate the seedling, tillering, jointing, flowering, and seed-filling stages, respectively. Data represent means ± SD (*n* = three biological replicates). Asterisks indicate statistical significance based on two-tailed Student’s *t* tests (* *p* < 0.05). FW: fresh weight.

## Data Availability

Data are available upon reasonable request from the corresponding author.

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
