# Peer review of "Decreased Nitrogen and Carbohydrate Metabolism Activity Leads to Grain Yield Reduction in Qingke Under Continuous Cropping"

_plants, 2025, doi:10.3390/plants14142235_

Round 1
Reviewer 1 Report
Comments and Suggestions for Authors
Title: Decreased Nitrogen and Carbohydrate Metabolism Activity Lead to Grain Yield Reduction in Qingke under Continuous Cropping
Summary: Ma et al. investigated the effects of 23 years of continuous cropping on nutrient availability for Hordeum vulgare growth and reported reduced levels of several key nutrients, including nitrogen (N), phosphorus (P), and potassium (K). However, the article exhibits several issues related to experimental inadequacies, a lack of novelty, and ethical concerns. Please find my detailed comments below.
- Over 30% of the content in the current manuscript appears to be duplicated from the authors’ previously published article titled “Elevated ROS Levels Caused by Reductions in GSH and AsA Contents Lead to Grain Yield Reduction in Qingke under Continuous Cropping” "https://doi.org/10.3390/ plants13071003”. This raises significant ethical concerns regarding self-plagiarism.
- Continuous cropping of Hordeum vulgare on the same land without rotation is expected to deplete soil nutrients, particularly because it is a non-leguminous crop and lacks the ability to rejuvenate soil through biological nitrogen fixation. As such, the study offers limited novelty, as the negative effects of continuous cropping under these conditions are well-documented.
- Given the regulatory roles of glutathione (GSH) and ascorbic acid (AsA) in nitrogen and carbohydrate metabolism under stress, primarily through redox signaling and maintaining metabolic homeostasis, it is unclear how the authors have integrated their current findings with their previously published work. In one instance, they report increased yield, while in another, they report decreased yield. This contradiction raises concerns about the consistency and interpretation of their results.
- The authors investigated a 23-year continuous cropping (CC) system but presented results only from the terminal year, leaving many critical questions unanswered. To fully understand the long-term effects, it is essential to report year-on-year trends in the measured parameters. Relying solely on terminal year data is insufficient to support the study’s conclusions.
- The article contains several grammatical issues, making it difficult to read.
Other concerns.
L38: "n biosynthesis—processes" Should this be hyphenated?
L39-41: This sentence unreadable. Please revise. Also, read the following articles to gain insight into nitrogen metabolism in plants and modify this section of the introduction (https://doi.org/10.1111/pce.14586 ; https://doi.org/10.1016/j.plantsci.2020.110759).
Line 44: "NR is the first enzyme………………..metabolism regulation." Please insert citation here
L46: Delete "It has been published" and rewrite the sentence
L49, 50: "level" "produced" "The function of", and many parts of the MS. Please avoid redundancy of words, and correct misplaced or misused words in sentences. This is widespread in the MS.
What is the trend of accumulation of N, P, K, NR, GS, etc., during this period?
L141: What does this mean, "The N, P and K leaves…?
L140: ……which declining from T1 ” Please rewrite this sentence
L160: The N (a), P (b), and K (c) content of continuous cropping of Qingke from Stages…
Content of what? Please be specific
L121: ……………use efficiency"?
L470: Name the specific part of the plant sample collected
L479….." the fresh samples were killed at 100 °C for….." How? "kill samples"
4.5.Enzyme activity Assays
4.4.Carbohydrate Profiling
4.3.Leaf Nutrient Content Analysis
4.2.Chlorophyll Content Quantification
Please provide detailed description of the above experimental pipeline

the writeup contain several grammatical inadequacies and redundant sentences.
Author Response
Dear reviewer, the response to your comments was listed in the attachment. Thank you for your constructive suggestions.

Reviewer 2 Report
Comments and Suggestions for Authors
Dear All,
The manuscript evaluates the impact of long-term continuous cropping (23 years) on Qingke (Hordeum vulgare L. var. nudum Hook. f.) through physiological, enzymatic, and metabolic lenses. The authors focus on nutrient depletion (N, P, K), photosynthetic pigment content, nitrogen/carbohydrate metabolism, and related enzymatic dynamics across five developmental stages. The findings emphasize that reductions in nutrient uptake and carbon metabolism, particularly in key enzymes such as GS, GOGAT, NR, SPS, and AGPase, are central contributors to grain yield decline under continuous cropping conditions.
This study addresses an important issue in sustainable Tibetan Plateau agriculture and provides novel insights into Qingke physiology under long-term monoculture. The biochemical and physiological metrics are detailed and follow a structured developmental timeline. However, key weaknesses concerning experimental transparency, biological interpretation, and linguistic clarity must be addressed to strengthen the manuscript’s scientific impact.
Strong Points
-
The study explores a chronically underreported challenge: physiological impairment in Qingke due to continuous cropping, using a 23-year field experiment, which adds substantial value to the literature.
-
The integrative approach - linking chlorophyll content, NPK dynamics, and carbohydrate metabolism with enzyme activities - provides a mechanistic view of yield decline.
-
References are well-suited and demonstrate a strong understanding of crop nitrogen and carbon metabolism.
Weaker Aspects
In my opinion, the critical handicapped points were:
-
Experimental Design Transparency: The manuscript lacks clear details about the physical and chemical properties of the soil, climatic conditions, irrigation status, field management practices, and experimental plot dimensions.
-
Biological Relevance and Hypothesis Framing: The manuscript does not define a central hypothesis. Additionally, the rationale for choosing Qingke and the significance of 23-year continuous cropping should be better justified.
-
Nutrient Deficiency vs. Enzymatic Decline Causality: While correlations are drawn between enzyme activity and nutrient content, causality is speculative. The manuscript should incorporate a more nuanced discussion on whether enzymatic suppression is a result or cause of nutrient deficits.
-
Carbon Starvation Evidence: Although starch and glucose dynamics are presented, the term “carbon starvation” is used without concrete evidence (e.g., photosynthesis rate, sugar transport impairment, or sink strength).
-
Enzyme Source and Method Limitations: Nearly all enzymatic data derive from commercial kits without descriptions of assay specificity, replication, or standardization controls. Enzyme isoform specificity (e.g., SS-I vs. SS-II) also lacks molecular validation.
-
Statistical Rigor and Replication: There is no mention of experimental unit definitions, blocking factors, or field heterogeneity. The study appears to have n=3, which may be insufficient for drawing strong conclusions from field trials with multi-year implications.
-
Autotoxicity and Microbiome Consideration: Though autotoxicity is mentioned in the discussion, there is no empirical assessment (e.g., root exudate profile or soil microbial community shifts). This remains speculative and disconnected from the presented data.
-
Language and Grammar: The manuscript contains numerous grammatical errors and unclear sentence constructions. These impede the clarity of scientific arguments (see examples below).
-
Inadequate Controls for Temporal Variability: Sampling was conducted across five growth stages, but without parallel environmental data (e.g., temperature, rainfall, light intensity) to explain physiological changes.
-
Conclusion Generalization: The suggestion to rotate crops is not supported by empirical or referenced evidence within the study. This weakens the applicability of the conclusion.
So, I have annotated below (and along the manuscript) an attempt to clarify certain ideas, but the authors should examine my suggested wording changes carefully to be sure that I have not misinterpreted what they wanted to say.
-
Page 1, Line 11 – "suffers from severe yield reduction under continuous cropping" → consider specifying the yield reduction percentage observed historically.
-
Page 1, Line 19 – "including glutamate synthase (GOGAT), glutamine synthetase (GS), and nitrate reductase (NR)" – introduce abbreviation before use.
-
Page 2, Line 42 – "aminotransferase" – specify which aminotransferase is being referred to (e.g., AAT or AST).
-
Page 3, Line 73 – "Plant sugars... serve dual roles" – should clarify that sugars act as osmolytes and metabolic signals.
-
Page 4, Line 123 – "followed by sharp declines of 6.2%, 90.98%..." – clarify what the two percentages refer to (are these sequential stage drops?).
-
Page 4, Line 131 – "but higher in the later stage" – define "later stage" numerically or temporally (T4–T5).
-
Page 5, Line 139 – "N, P and K leaves" – change to “leaf N, P, and K content.”
-
Page 6, Line 168 – "progressive GOGAT activity declines" – include specific values or fold-change to contextualize.
-
Page 6, Line 173 – "photorespiration" – mention whether this was assessed directly.
-
Page 7, Line 187 – "undetected at T4 and T5" – define method detection limit.
-
Page 8, Line 231 – “1.81-fold difference” - clarify which group is higher.
-
Page 9, Line 278 – "suggesting stage-specific inhibit or accumulation" - grammatically incorrect; rephrase.
-
Page 10, Line 299 – "distinct kinetic characteristics" - vague; please clarify.
-
Page 11, Line 337 – "pyrophosphate: fructose-6-phosphate 1-phosphotransferase (PFP)" - abbreviation used inconsistently.
-
Page 12, Line 386 – "significantly lower than that of the control group" - add actual values or statistical metrics.
-
Page 13, Line 419 – "resultant carbon starvation" - not validated with direct photosynthetic or phloem transport data.
-
Page 13, Line 425 – “irreversible yield penalty cascade” - overly speculative.
-
Page 13, Line 426 – “molecular mechanisms…inadequately resolved” - vague. What specific molecular gaps remain?
-
Page 14, Line 451 – "straight chain alcohols, fatty aldehydes..." - inconsistent list format and unclear if these are empirically verified in Qingke.
-
Page 14, Line 462 – “autotoxic feedback loops” - speculative, lacks chemical quantification in the study.
-
Page 15, Line 471 – "randomized complete block design" - missing field layout details.
-
Page 15, Line 474 – "fertilizer was applied at a ratio of 22:10:13" - indicate actual units (kg ha⁻¹).
-
Page 15, Line 475 – "no irrigation" - how was drought stress controlled or monitored?
-
Page 15, Line 478 – “fresh samples were killed at 100 °C” - inappropriate phrasing, rephrase to “inactivated.”
-
Page 16, Line 487 – "molybdenum blue method" - provide citation or reference.
-
Page 17, Line 497 – “significance analysis with Duncan’s multiple range test” - consider ANOVA + post-hoc.
-
Page 18, Line 506 – "photo-assimilates deposition" – unclear phrasing, use “accumulation and partitioning.”
-
Page 18, Line 510 – "sucrose and starch contents accumulated lower" - grammatical issue; rephrase.
-
Page 18, Line 515 – “poor performance of plant growth” - vague; use more precise terminology.
-
Page 18, Line 517 – “rotation with different crops” - please suggest specific rotation crops suitable to Qingke systems.

Author Response
Dear reviewer, the responses to your comments were listed in the attachment. Please confirm it. Thank you for your constructive suggestions.

Reviewer 3 Report
Comments and Suggestions for Authors
This manuscript investigates the physiological and biochemical mechanisms underlying grain yield reduction in Qingke (Hordeum vulgare L. var. nudum) subjected to 23 years of continuous cropping. The study assesses chlorophyll content, macronutrient levels (N, P, K), activities of key nitrogen- and carbohydrate-metabolizing enzymes, and sugar and starch contents across five developmental stages. The findings suggest that long-term monoculture impairs nitrogen assimilation and carbon allocation, contributing to yield loss.
The study addresses an important agronomic problem of continuous cropping stress in Qingke, a crop critical to the Tibetan Plateau. The time-course sampling across developmental stages provides insights into dynamic metabolic changes. A comprehensive set of physiological, enzymatic, and biochemical parameters is assessed to support conclusions.
The manuscript provides useful data on continuous cropping effects in Qingke but requires substantial improvements in interpretation, writing quality, statistical rigor, and integration of findings into a coherent framework.
Major Comments
- Conceptual Integration Lacking
While the results are extensive, the manuscript lacks a cohesive narrative tying them to a central mechanistic model. The data are presented as a collection of trends without a synthesis that builds explanatory insight into how nitrogen and carbohydrate imbalances drive yield loss.
The Discussion should be restructured around key mechanistic pathways (e.g., impaired nitrogen assimilation → reduced carbon sink strength → yield loss) to improve interpretability.
2. Statistical Rigor and Replication
Replication details are insufficient. While the manuscript states n = 3, there is no indication of whether these are independent biological replicates or pseudoreplications (e.g., pooled samples).
Clearly define the biological and technical replicates and justify sample size. Add test statistics (e.g., F-values, p-values) in figure legends or supplementary data.
3. Language and Grammar
The manuscript suffers from awkward phrasing and frequent grammatical errors that hinder readability. Examples include:
“The N, P and K leaves…” → should be “The N, P, and K contents in leaves…”
“Data was observed by ANOVA” → grammatically incorrect; revise to “Data were analyzed using ANOVA.”
A thorough professional language revision is essential to improve clarity and precision.
4. Enzyme Activity Interpretation
The discussion of enzyme activity (e.g., GOGAT, SPS, PPDK) lacks mechanistic depth. The biological implications of altered enzymatic patterns are not adequately explained, particularly how they contribute to nitrogen use efficiency or carbohydrate partitioning.
Discuss the metabolic consequences of reduced enzyme activity and relate to known regulatory networks in cereal crops.
5. Literature Context
The manuscript cites older or general references but could benefit from more targeted comparisons with recent studies on monoculture-induced stress or nitrogen/carbon interactions in cereals.
Include more recent studies in barley, wheat, or maize addressing nutrient stress and enzyme regulation under similar conditions.
6. Conclusion and Future Directions
The conclusion reiterates results without proposing practical strategies or future research directions.
Provide a more critical synthesis of findings and propose testable hypotheses or agronomic interventions (e.g., crop rotation, microbial inoculants, fertilizer optimization).
Minor Comments
Define all abbreviations at first mention (e.g., AGPase, TPI, SPS).
Ensure consistent formatting of gene/enzyme names (italicize gene names, not proteins).
Units and scales in figures need standardization (e.g., µg/g FW, nmol/min/mg protein).
Check for consistency in terminology (e.g., “23y-CC” vs. “continuous cropping treatment”).
Author Response

(The authors gave the same response as above.)

Round 2
Reviewer 1 Report
Comments and Suggestions for Authors
Although the authors have addressed some of my concerns, their failure to indicate line numbers showing where those comments were addressed has made the revision of this manuscript quite difficult. Nonetheless, there are still a few issues that require clarification in the manuscript.
- Authors response to my fourth comment showed that current report was not based on the 23 year CC, as suggested. It is however, imperative to categorically state at which point the current study reported these results.
- Please incorporate the response to the comments below in the revise MS."
What is the trend of accumulation of N, P, K, NR, GS, etc., during this period?"
- Line 39 "which is called,..........." Please delete, not suitable in the context
- Lines 125-127: Grammatical infraction. please revise
- Line 131: Separate the nutrient by a comma and and not a hyphen
- Line 269-270: Please rephrase this sentence
English language must be edited.
Author Response
First of all, we would like to express our appreciations to you. We also made quantity of revisions to improve our manuscript quality based on your comments. English was also corrected by MDPI English service.
Comments and Suggestions for Authors
Although the authors have addressed some of my concerns, their failure to indicate line numbers showing where those comments were addressed has made the revision of this manuscript quite difficult. Nonetheless, there are still a few issues that require clarification in the manuscript.
Response: We are very sorry that we do not use track or line numbers to indicate where had been made changes clearly. We upload the original and revised version versions at this time. Because we made quantity of revisions, we used track system to show where we have made revisions. Thank you for your suggestions.
- Authors response to my fourth comment showed that current report was not based on the 23 year CC, as suggested. It is however, imperative to categorically state at which point the current study reported these results.
Response: Thanks. We appreciate this important comment regarding experimental design. Our analysis specifically focuses on the control group and 23y-CC treatment which represent the mildest and the most severe levels of continuous cropping barrier respectively. We also measured the Chlorophyll, starch, sucrose content and other physiological indexes and found that the contents were significantly higher in the 5y-CC treatment as compared to the control. These results suggest that shorter durations of continuous cropping have a minimal impact on Qingke. In our previous study, we also analyzed ROS levels in the control group and the 23y-CC treatment, which showed a significant decrease in Qingke grain yield [43]. Based on these findings, we selected the control group and 23y-CC treatment in this study.
We sincerely apologize for not being able to include continuous time points in the current study due to the lack of mid-term samples in our experimental design. Although these additional data could provide further insights, the statistically robust trends observed between the two endpoints (control and 23y-CC) remain valid. I hope the reviewer can understand us. Thank you.
[43]
- Please incorporate the response to the comments below in the revise MS."
What is the trend of accumulation of N, P, K, NR, GS, etc., during this period?"
Response: Thanks for your suggestion. We have incorporated this part to the manuscripts. Line 172-173; line 222; line 241-242.
- Line 39 "which is called,..........." Please delete, not suitable in the context
Response: Thanks for your valuable suggestion. We have rewritten this sentence. We are sorry for the grammatical errors and awkward constructions, and our manuscript has been professionally edited to enhance readability.
- Lines 125-127: Grammatical infraction. please revise
Response: Thanks for your valuable suggestion. We have rewritten this sentence.
- Line 131: Separate the nutrient by a comma and and not a hyphen
Response: Thanks for the scientific suggestions. We have revised it
- Line 269-270: Please rephrase this sentence
Response: Thanks for your valuable suggestion. We have rewritten this sentence.

Reviewer 3 Report
Comments and Suggestions for Authors
- Language and Style (Still Needs Work)
While the revised manuscript is improved, English usage remains uneven, with persistent grammatical errors and awkward constructions.
Examples:
“The N, P, and K leaves content decreased…” → should be “The N, P, and K contents in leaves decreased…”
“...could break the carbon and nitrogen metabolism balance” → consider “...could disrupt the balance between carbon and nitrogen metabolism.”
Recommendation: Proofreading.
- Figure Formatting and Consistency
While units have been standardized, the visual quality of some figures (e.g., enzyme activity bar plots) remains suboptimal. Axis labels and legends could be clearer. Font size is small in some cases.
Recommendation:
Increase label and axis font sizes for legibility.
Explicitly state sample size (n = 3) and significance annotations in each figure legend.
Ensure consistency in the format of units (e.g., “mg/g FW,” “nmol/min/mg protein”).
- Depth of Mechanistic Interpretation
The discussion has improved but remains somewhat descriptive. The roles of key enzymes (e.g., GOGAT, AGPase, SPS, PPDK) could be more deeply tied to known pathways in nitrogen use efficiency and carbon sink strength in cereals.
Recommendation: Expand on how enzyme activities affect:
Source–sink relations during grain filling.
Known nitrogen signaling pathways in barley or related species.
- Ambiguity in Terminology and Abbreviations
The use of “23y-CC” is now consistent, but some abbreviations (e.g., TPI, PK) are still not defined at first mention or appear abruptly.
Author Response
Comments and Suggestions for Authors
- Language and Style (Still Needs Work)
While the revised manuscript is improved, English usage remains uneven, with persistent grammatical errors and awkward constructions.
Response: Very thank you for your concerns. We have corrected English as recommended by MDPI English service.
Examples:
“The N, P, and K leaves content decreased…” → should be “The N, P, and K contents in leaves decreased…”
Response: Thanks for your valuable suggestion. We have rewritten this sentence.
“...could break the carbon and nitrogen metabolism balance” → consider “...could disrupt the balance between carbon and nitrogen metabolism.”
Response: Thanks for your valuable suggestion. We have rewritten this sentence.
Recommendation: Proofreading.
Response: Thanks for your suggestion. Our manuscript has been professionally edited to enhance readability.
- Figure Formatting and Consistency
While units have been standardized, the visual quality of some figures (e.g., enzyme activity bar plots) remains suboptimal. Axis labels and legends could be clearer. Font size is small in some cases.
Recommendation:
Increase label and axis font sizes for legibility.
Explicitly state sample size (n = 3) and significance annotations in each figure legend.
Ensure consistency in the format of units (e.g., “mg/g FW,” “nmol/min/mg protein”).
Response: Thanks for your valuable suggestion. We have now made the following revisions to address these concerns:
- Increased font sizes of all axis labels and legends for better readability. The modified figures (Figures 1-6) have been updated in the revised manuscript.
2.Annotation ‘Data represent means ± SD (n = 3 biological replicates). Asterisks indicate statistical significance by two-tailed Student’s t tests (*P < 0.05).’ in each figure legend.
3.Standardize unit formatting: use 'mg/g' for all content and 'nmol/min/g FW' for all enzyme measurements.
Depth of Mechanistic Interpretation
The discussion has improved but remains somewhat descriptive. The roles of key enzymes (e.g., GOGAT, AGPase, SPS, PPDK) could be more deeply tied to known pathways in nitrogen use efficiency and carbon sink strength in cereals.
Recommendation: Expand on how enzyme activities affect:
Source–sink relations during grain filling.
Known nitrogen signaling pathways in barley or related species.
Response:We sincerely appreciate your insightful suggestion to deepen the mechanistic interpretation of enzyme functions in nitrogen-carbon coordination. We have added some samples to explain how enzyme affect source–sink relations during grain filling. In rice, OsWRKY23-DNR1 module enhances nitrate uptake and GOGAT activity, boosting NUE in indica vs. japonica subspecies [50]. NR activity increases under shading condition but exacerbates C-N imbalance, reducing grain quality [51]. AGPase rate-limiting enzyme for starch biosynthesis, low temperatures reduce AGPase activity in wheat, decreasing amylose/amylopectin content [52]. In Line 490-496.
[50] Zhang, S.Y.; Ji, Z.; Jiao, W.; Shen, C.B.; Qin, Y.J.; Huang, Y.Z.; Huang, M.H.; Kang, S.M.; Liu, X.; Li, S.Q.; Mo, Z.L.; Yu, Y.; Jiang, B.Y.; Tian, Y.A.; Wang, L.F.; Song, Q.X.; Wang, S.K.; Li, S. Natural variation of OsWRKY23 drives difference in nitrate use ef-ficiency between indica and japonica rice. Nat. Commun. 2025, 16, 1420.
[51] Chen, X.Y.; Zhu, Y.; Ma, Z.T.; Zhang, M.Y.; Wei, H.Y.; Zhang, H.C.; Liu, G.D.; Hu, Q.; Li, G.Y.; Xu, F.F. Effects of light inten-sity and nitrogen fertilizer interaction on carbon and nitrogen metabolism at grain-filling stage and its relationship with yield and quality of southern soft japonica rice. Acta Agronomica Sinica. 2023, 49, 3042-3062.
[52] Liu, J.M.; Si, Z.Y.; Li, S.; Wu, L.F.; Zhang, Y.Y.; Wu, X.L.; Cao, H.; Gao, Y.; Duan, A.W. Effects of water and nitrogen rate on grain-filling characteristics under high-low seedbed cultivation in winter wheat. J Integr Agr. 2023, 23, 4018-4031.
- Ambiguity in Terminology and Abbreviations
The use of “23y-CC” is now consistent, but some abbreviations (e.g., TPI, PK) are still not defined at first mention or appear abruptly.
Response: Thanks for the scientific suggestion. We have now defined all abbreviations at first use in the main text: ‘Glutaminase (GLS)’ in Line 227; ‘Sucrose phosphorylase (SP)’ in Line 342; ‘Pyrophosphate: Fructose-6-phosphate 1-phosphotransferase (PFP)’ in Line 394; ‘Fructose-1.6-diphosphatase (FBP)’ in Line 407; ‘Pyruvate phosphate dual kinase (PPDK)’ in Line 429; ‘triose phosphate isomerase (TPI)’ in Line 441.
